# Promote Community Engagement in Participatory Research for Improving Breast Cancer Prevention: The P.I.N.K. Study Framework

**DOI:** 10.3390/cancers14235801

**Published:** 2022-11-25

**Authors:** Michela Franchini, Stefania Pieroni, Francesca Denoth, Marco Scalese Urciuoli, Emanuela Colasante, Massimiliano Salvatori, Giada Anastasi, Cinzia Katia Frontignano, Elena Dogliotti, Sofia Vidali, Edgardo Montrucchio, Sabrina Molinaro, Tommaso Susini, Jacopo Nori Cucchiari

**Affiliations:** 1Institute of Clinical Physiology, National Research Council of Italy, 56124 Pisa, Italy; 2Senologica Srl, 19124 La Spezia, Italy; 3Umberto Veronesi Foundation, 20122 Milano, Italy; 4Diagnostic Senology Unit, Azienda Ospedaliero Universitaria Careggi, 50139 Firenze, Italy; 5Breast Unit, Gynecology Section, Department of Health Sciences, University of Florence, 50139 Firenze, Italy

**Keywords:** breast cancer, lifestyle, risk profile, data-driven approach, social network analysis, prevention, health literacy, health education, self-empowerment, mobile health tools

## Abstract

**Simple Summary:**

More than 50% of breast cancers may be preventable with adherence to healthy lifestyle practices, but the influences of each single preventive/predisposing behaviour and the effects of their combination are still widely debated. The aim of our study was to identify combinations of non-modifiable and lifestyle-related factors that could influence the chance of having breast cancer in post-menopausal women. We used a twofold strategy of analysis that combines traditional statistical methods and innovative data-driven approaches. We identified some combination of women’s features and habits at higher risk for breast cancer occurrence. These preliminary findings could be used to inform tailored prevention policy and health education programs for improving communities’ self-empowerment.

**Abstract:**

Breast cancer (BC) has overtaken lung cancer as the most common cancer in the world and the projected incidence rates show a further increase. Early detection through population screening remains the cornerstone of BC control, but a progressive change from early diagnosis only-based to a personalized preventive and risk-reducing approach is widely debated. Risk-stratification models, which also include personal lifestyle risk factors, are under evaluation, although the documentation burden to gather population-based data is relevant and traditional data collection methods show some limitations. This paper provides the preliminary results from the analysis of clinical data provided by radiologists and lifestyle data collected using self-administered questionnaires from 5601 post-menopausal women. The weight of the combinations of women’s personal features and lifestyle habits on the BC risk were estimated by combining a model-driven and a data-driven approach to analysis. The weight of each factor on cancer occurrence was assessed using a logistic model. Additionally, communities of women sharing common features were identified and combined in risk profiles using social network analysis techniques. Our results suggest that preventive programs focused on increasing physical activity should be widely promoted, in particular among the oldest women. Additionally, current findings suggest that pregnancy, breast-feeding, salt limitation, and oral contraception use could have different effects on cancer risk, based on the overall woman’s risk profile. To overcome the limitations of our data, this work also introduces a mobile health tool, the Dress-PINK, designed to collect real patients’ data in an innovative way for improving women’s response rate, data accuracy, and completeness as well as the timeliness of data availability. Finally, the tool provides tailored prevention messages to promote critical consciousness, critical thinking, and increased health literacy among the general population.

## 1. Introduction

The World Health Organization announced that breast cancer (BC) has overtaken lung cancer as the most common cancer in the world [1]. BC is one of the most frequently diagnosed cancers in women with 2.26 million new cases in 2020, and it is the leading cause of cancer death in women worldwide, with an age-adjusted rate of 13.6/100,000 [2]. Current projections indicate that by 2030 the worldwide number of newly diagnosed cases will reach 2.7 million annually, also as the consequence of current ageing population [2]. It is widely proven that screening for BC reduces deaths from cancer [3]. Most countries use age-based or “one-size-fits-all” breast screening approaches, which do not consider the wide variation in individual BC risks [4], although the cost-effectiveness and the benefit-to-harm ratio of BC programs could be improved by adopting a risk-stratified screening strategy [4,5].

The general hypothesis is that comprehensive models of risk-stratification could affect screening intensity/interval, starting age, imaging modality use, or even decisions not to screen [6]. The emergent pattern states that the incorporation of multiple datasets from clinical assessment (breast density, other mammographic features, and indexes from ultrasound or tomographic methodologies) as well as from patients’ lived experience (first-degree family history of BC, occupational exposures, increased body mass index (BMI), nulliparity, or young age at first birth) or genetic information, yields incremental gains in risk model performance [7,8].

To date, the efficacy and feasibility of personalising screening strategies is still uncertain, albeit a wide range of risk models to quantify the combined effect of numerous BC risk factors, have been developed for clinical use [9,10,11]. Many risk models have been validated in study populations other than those used in the initial development or have been further assessed in comparative studies. Risk prediction models generally consider well-recognised ‘risk factors’ (breast density, first-degree family history of BC, increased body mass index (BMI), nulliparity, or young age at first birth) and other personal features that influence BC risk [11].

Modifiable BC risk factors include radiation exposure, hormone replacement therapy, alcohol and high fat diet, smoking, and exposure to some environmental factors such as organochlorine chemicals and electromagnetic fields [12]. Breast tissues of nulliparous and never-breastfed women are more likely to mutate and develop BC. Mammary cells differentiation completely occurs after pregnancy and during the lactation period and undifferentiated cells are more likely to be susceptible to carcinogenic substances [13]. Moreover, levels of circulating estrogens and androgens are associated with BC risk in both premenopausal and postmenopausal women [14,15]. The biology of progesterone receptors in the normal mammary gland and in BC risk provides a framework for understanding how chemicals that affect hormone homeostasis may alter breast development and ultimately cancer risk. Lower cumulative exposure to estrogen seems to protect against BC [16], while higher exposure to progesterone may increase risk of BC, and affecting progesterone or progesterone receptor signaling pathways promotes BC progression [17,18]. BC risk is also elevated by extended exposure to high levels of endogenous hormones, which can occur with obesity [19] or as a result of early age at menarche or late age of first pregnancy and menopause [20]. Furthermore, an exposure to compounds that have estrogen-like activity has been shown to influence normal mammary development and lead to adverse lifelong consequences, especially when exposures occur during early life [21].

Additionally, existing evidence suggests that up to 50% of BC cases may be preventable by adherence to healthy lifestyle practices. In particular studies of a cohort of both pre- and postmenopausal women showed 31% lower rates of BC in women who adhered to specific dietary recommendations of increasing wholegrain products and reducing meat and alcohol, rather than other lifestyle factors. Both the American Cancer Society (ACS) Guidelines and the World Cancer Research Fund/American Institute for Cancer Research (WCRF/AICR) Recommendations suggest maintaining a healthy weight throughout life, consuming a plant-based diet, adopting a physically active lifestyle, and limiting red meat and alcohol consumption [22,23]. The WCRF/AICR guidelines include legumes and grains within their vegetable and fruit recommendation, and suggest limiting energy density and sodium intake. The ACS guidelines include a recommendation to choose whole grains [24]. Furthermore, the Third Expert Report published by the World Cancer Research Fund (WCRF) concluded that there is strong evidence that vigorous physical activity protects against premenopausal BC, moderate or vigorous physical activity protects against postmenopausal BC, and greater body fatness and weight gain in adult life causes postmenopausal BC [22,23].

Alcohol intake increases serum estrogen levels, possibly by stimulating aromatase activity, and alcohol has been consistently linked to an increased risk of developing BC. Additionally, maternal alcohol consumption during pregnancy increases female offspring’s mammary tumorigenesis as well as a maternal intake of a high-fat n-6 polyunsaturated fatty acid diet [25]. Hyperinsulinemia and type 2 diabetes were shown to be independent risk factors for postmenopausal BC, too [26].

A recent study about the burden of disease from BC attributable to direct and indirect tobacco smoking exposure reported that smoking accounts for 2.6% of DALYs lost and second-hand smoke for 1.0% [27]. Additionally, Pizzato et al. [28] estimated that the risk of androgen receptor-positive BC onset among ever-smokers, compared to never smokers, is three times higher (OR: 2.85 (95% CI 1.02–7.96)).

In general, healthy eating patterns and healthy behavior such as keeping physically active are widely recognized as potentially being more important for chronic disease prevention than the intake or exclusion of specific food items or nutrients [29].

The multifactorial assessment of BC risk, also using data points having ‘weaker’ effects on risk, makes the long-term prediction of BC particularly challenging and requires the availability of large amounts of data on the entire population [11].

Physicians report spending 34 to 37% of their time on data documentation and processing to implement Electronic Health Records, and they would ideally spend less time on this task [30], as the documentation burden and time were significantly associated with a decrease in dedicated time for patient care. A recent study [31] underlaid the lack of recommendations to perform regular physical activity from the oncology clinicians, despite the evidence that exercise is associated with a lower risk of developing cancer and improved survival after a cancer diagnosis, in particular among patients with breast, colon, and prostate cancer. Observed barriers to clinicians referring patients to exercise programming include the belief that it is not within the scope of practice for oncology clinicians. The lack of systems for triage incorporating physical activity assessment as a standard part of chronic patient health care is a further limitation [31,32,33].

The need for innovative approaches and tools to collect population-based data about more specific personal features or behavioural risk factors in a more accurate way is widely reported [32].

This also agrees with the conclusions reported in the Procas studies (Procas 1 and Procas 2), involving more than 50.000 women invited for breast screening, aimed at combining lifestyle, reproductive history and other clinical information with imaging assessment of mammographic density and DNA obtained for polygenic risk analysis [34].

In particular, risk information was collected using a two-page questionnaire mailed to women in the interval between receiving the call for screening and attendance for calculating women’ personal 10-year BC risk, according to the Tyrer–Cuzick (TC) model. The TC model, which is used to identify women at moderate- and at high-risk level for BC, is based on the combination of extensive family history, endogenous oestrogen exposure, and benign breast disease (atypical hyperplasia). Notwithstanding the large population involved in the studies, the Procas’ researchers reported some common limitations affecting self-reported data. One is the low participation (43%) in the study by the screened population and the second is the inaccuracies in women’s filling-in of the questionnaires. The authors concluded that using an online version of the questionnaire is likely to improve accuracy in collecting data.

Our hypothesis is that the use of electronic or mobile tools [35] together with a community-based participatory research approach could help to overcome these limitations [32,33,36].

Within the past few decades, patients have become increasingly active in their own medical care and participating patients define themselves as collaborating actors who are able to state their preferences in the process of decision-making [37,38]. Additionally, community-based participatory research focusing on research, action, and education allows people to face cancer prevention and control challenges [39]. Participatory approaches offer opportunities to integrate knowledge held by stakeholders (community members, patients, caregivers, etc.) into the formal scientific literature and to influence the evidence-based guidelines, as stakeholders and researchers collaborate to build real-world evidence [40]. A further hallmark of the participatory approach consists of enhancing patients’/communities’ capacity to handle current and future health challenges and to discern between good-quality scientific evidence and “junk science” [41]. An individual ability to engage with data and research evidence allows people to react to the dynamic cancer prevention and control evidence base, whether new evidence-based practices are implemented or those that are no longer the standard of care are dropped. Many studies showed that health literacy (HL), defined as the knowledge, motivation, and competence to access, understand, appraise, and apply health information, is predictive for individual health behaviour, and it is an important prerequisite for patient participation in healthcare [42].

Notwithstanding that research findings derived from randomized controlled trials are considered the gold standard for effectiveness research, evidence from real-world data offers enormous opportunity (a) for understanding the cumulative effects of cancer risk factors more deeply, (b) evaluating the efficacy and limits of treatments among different subgroups of people, and (c) advancing a patient-centered health ecosystem [40].

The Prevention, Imaging, Network, Knowledge (P.I.N.K.) study is an on-going seven-year longitudinal multicenter study, which was started in October 2017, aiming to recruit 50,000 women of the age of 40 years and above, presenting spontaneously for routine breast examination at several public or private diagnostic centers across Italy. It is coordinated by the Italian National Research Council (CNR), co-funded by the Umberto Veronesi Foundation, and conducted through the collaboration of many Italian diagnostic centers performing clinical BC diagnosis [43]. The P.I.N.K. study collects data using self-administered paper questionnaires to investigate women’s risk factors, family history, and lifestyle in order to create a comprehensive set of information for enabling the definition of personalized risk profiles [43], with the final aim of promoting primary prevention. Additionally, the questionnaire data is linked to a database containing standardized clinical data collected by radiologists about the same women, to estimate the gained diagnostic accuracy of each diagnostic path, also considering the woman’s features.

In this work, we show the preliminary results and the methodological approach (model driven and data driven) used to analyze the P.I.N.K. [43] data to the aim of combining women’s personal features and lifestyle habits into profiles that could influence the BC risk. To overcome the limitations of self-reported data collected using a traditional approach (paper questionnaire), our work also aims to introduce an innovative tool, the Dress-PINK, designed to collect population-based information in the BC field.

## 2. Materials and Methods

### 2.1. The PINK Study Data and the Study Population

To date, out of all the women who have presented spontaneously for routine breast examination at the sixteen P.I.N.K. diagnostic centers in Italy, 29,355 women decided to participate in the study. The women presenting at the centers for their visits all received a clinical breast examination and underwent a diagnostic examination. The diagnostic path includes a mammography and at least a second instrumental test among digital breast tomosynthesis (DBT), ultrasonography (US), magnetic resonance (MRI), and contrast enhanced spectral mammography (CESM) [43]. The mammography is directly performed within a P.I.N.K center, or it is subjected to a second reading if a woman attends the center to integrate further instrumental tests to a mammography recently performed within the national BC screening. The enrolled women are also invited to fill a structured self-administered questionnaire investigating their risk factors, family history, and lifestyle through a number of items organized into four main sections: (a) social characteristics such as marital status, education level, and job situation; (b) anamnesis (age of menarche and menopause, hormonal therapies, oral contraception, pregnancy and breastfeeding, and assisted fertilization); (c) lifestyle, eating habits, physical exercise, smoking habits, and alcohol consumption; and (d) the family history of detecting the presence of tumors among first degree relatives.

The P.I.N.K. questionnaire items were jointly decided by the group of radiologists and epidemiologists participating in the study based on a Delphi method, and the questionnaire comprehension was tested among a random sample of 50 women of the general population.

During the study, the diagnostic centers collected the completed questionnaires and returned them to the CNR, where the questionnaires were processed with an optical reading technique. The radiologists participating in the P.I.N.K. study also collected clinical data (personal information, clinical breast examination, MX, DBT, US, MRI, CESM, cytological or micro-histological reports, diagnostic conclusions, and cancer cases in-depth description) using a web platform specifically developed for this purpose.

At each subsequent access by the same woman, the clinical information was updated by radiologists directly on the web platform, while the contextual information was provided by the PINK women through the self-administration of a short follow-up questionnaire. The questionnaire digital archives resulting from the optical reading were integrated with the clinical data entered in the PINK web platform by the clinicians. To date, data linking was performed for 10,097 women (55.5% in postmenopausal status).

The preliminary results presented in this work concern 5601 postmenopausal women. Among the premenopausal women, the BC cases were only 57, so we decided to exclude them from the study as the application of statistical analyses couldn’t provide stable results.

Regarding the 5601 postmenopausal women, they underwent more than one breast examination between the years 2018 and 2022 and filled out the baseline questionnaire. In order to investigate the risk factors potentially associated with the likelihood of having BC cancer or the possible confounding factors, many variables were analyzed. The output measure i.e. the BC diagnoses provided by radiologists, has been defined as a dichotomous variable (yes/not) using the anatomic pathology reports of BI-RADS 5 (B5) lesions resulting from needle core biopsies. To date, the final reports of the histological test after surgery for BC are available for 85% of the cases only, but a preliminary analysis about the concordance between all the anatomic pathology reports from the core biopsy and the surgery reports estimated a 97% compliance in identifying malignant lesions.

Each woman has been characterized on three different profiles based on self-reported and on radiologist collected data as well: non-modifiable factors (profile 1), personal history features (profile 2), and lifestyle habits (profile 3) at the baseline. Non-modifiable factors include age, breast density (four BI-RADS density levels indicated by radiologists), reproductive period length (four classes of time identified as the distribution’s quartiles), menopause status length (less than six years and at least five years) and family history (no first line relatives, female relatives, male relatives, or male and female relatives with BC). Personal history features include qualification (primary/secondary education, high school, university/post university education), occupation (unemployed, employed, retired, working night shifts), current or past co-morbidities (none, one co-morbidity, more than one comorbidity), pregnancy (yes/not), breastfeeding (yes/not), chest radiation therapy (CRT: yes/not), hormone use for contraception (never or less than one year, from one to five years, more than five years), ovarian stimulation or replacement therapy (HRT: yes/not), body mass index (BMI; underweight, normal weight, overweight, obese), and weight gain during menopause (yes/not). Lifestyle risk factors concern current and past smoking habits (no-smoker, low, medium, and high level); current alcohol drinking habit (no-drinker, low, medium, and high level); the compliance with the WCRF’s recommendations about improving physical activity and plant foods use, limiting energy-dense foods, red meat, and salt use; and following a varied diet (low, medium and high compliance).

### 2.2. Traditional Statistical Analysis and Network Implementation

The descriptive analysis of the women population was conducted by calculating the proportion of all the variables considered in our study. A binomial logistic regression model was also performed including all the variables that were statistically significant (*p* < 0.05) within the logistic regression univariate analyses (Figure 1). Each univariate analysis included age and a single variable of the P.I.N.K. questionnaire or the breast density.

The statistical analysis was carried out through SPSS Statistic 26, and the results are expressed as an Odds Ratio (OR) and 95% confidence interval.

The goal of this study was also to classify or stratify women using algorithms that are objectively data-driven (i.e., identify women groups based on similarity of their features inside each profile, with no pre-determination of how many women groups, how many variables were needed to define these groups, whether each group reflected different values of the same variables, or even if the same set of variables was used to define the individual groups). Therefore, each profile was analyzed further using the Social Network Analysis (SNA) techniques (Figure 1), based on the solid theoretical structure of the graph theory. The actual analysis is carried out using the community detection (CD) algorithm implemented by Gephi [44], an open source software platform that allows interactive exploration and analysis of complex networks.

Original raw data has been categorized before loading it into Gephi. We developed categorical boundaries (i.e., cut points) for each variable based on current standards and on clinicians input as potentially relevant to the study population. These boundaries also reflect, where appropriate, a high/normal/low classification. The result is a number of categories that defined individual characteristic nodes in the network; for example, BMI is defined in four categories: Underweight (<18.5), Normal weight (18.5–24.9), Overweight (25.0–29.9), and Obese (≥30.0). The data file loaded into Gephi for each of the three profiles generates an undirected graph: a node for every woman, a node for every characteristic, and an edge to connect each woman’s node to all its characteristic nodes. The characteristic nodes are connected to one another whenever they refer to the same woman. We generated three networks and three graphs for each profile population of women. In terms of overall network measure, the modularity class indicates how well a network decomposes into modular communities. One of the most widely accepted definitions indicates a community as “a group of nodes of a graph which are more strongly connected to each other than with other nodes in the same graph” [44]. The modularity is a scalar value between −1 and 1 that measures the density of links within communities as compared to links between communities.

Once the communities of each profile have been identified, each woman was characterized based on the combination she belongs to within the three profiles. Then, the combinations of communities were grouped into fifteen families of combinations sharing non-modifiable factors (profile 1) and personal history features (profile 2). In order to isolate the families that were statistically associated with the BC rate, we performed a monovariate binomial logistic regression for each family and isolated those families having a statistically significant OR at the 0.05 significance level.

The isolated families were compared with each other and with the group of women not belonging to the families considered, based on the distribution of the characteristics included in profile 1, 2, and 3 (Figure 1). The comparison was performed using the chi-square test and the standardized adjusted Pearson residual analysis [45].

## 3. Results

### 3.1. Population Characteristics

Postmenopausal women amounted to 5601 (mean age 59.7 ± 7.6 years) and, on average, they have reached menopause at least 9 years (9.5 ± 7.7 years) previously. About 73.0% of women showed BI-RADS B and BI-RADS C breast density (Table 1—“ALL women” column). The main language spoken by the family of origin was Italian (97.0%), followed by English/German language (1.0%). More than 39.0% have attained high school graduation (vs. 38.0% among the Italian women in 2019, aged 55–64 years [46]), while about 35.0% (vs. 12.8% among the Italian women in 2019, aged 55–64 years [46]) had university or post graduate degree. For the most part, women were employed (58.2% vs. 44.5% among the Italian women in 2019, aged 55–64 years [46]), while 39.3% were unemployed, housewives, or retired (Table 2—“ALL women” column). Among employed or retired women, 1.5% work/worked night shifts (Table 2—“ALL women” column).

A quarter of women reported a history of first female line relatives with BC (Table 1—“ALL women” column), 28.3% had at least one comorbidity and more than 58.0% had at least two co-morbidities, in particular cardiometabolic disease (58.6%), endocrine disease (50.9%), and neuropsychiatric diseases (41.2%). More than 50% used oral contraceptives for at least one year, while the large majority did not use hormone therapy for ovarian stimulation (93.1%) or hormone replacement therapy (81.6%). Pregnancy and breast feeding have been experienced by 80.3% and 64.1% of women, respectively. Only 8.2% of women underwent chest radiation therapy during their life (Table 2—“ALL women” column).

Regarding the lifestyle habits (Table 3—“ALL women” column), postmenopausal women reported low frequency of tobacco and alcohol use (26.0% smoke more than six cigarettes/day and 26.2% drink alcohol several times per week), a medium level of compliance with WCRF’s recommendations (about 40% show high/very high compliance) except for physical activity (high compliance 28.5%).

Among postmenopausal women, the BC cases detected by radiologists and confirmed by the cytological or micro-histological reports amounted to 100 (prevalence rate: 1.8%; 95%CI: 1.5–2.2%) and, on average, women were followed for 1.5 years.

### 3.2. Binomial Logistic Model

The final binomial logistic regression model was performed including all the variables that were statistically significant (*p* < 0.05) within the univariate analyses. According to the final model (Table 4), age (OR: 1.1, *p* = 0.00) and BI-RADS D density (OR: 2.5, *p* = 0.03), the most extreme dense breast tissue, were positively associated with BC as well as having first line male relatives with BC (OR: 5.5, *p* = 0.03). A high compliance with the WCRF’s recommendations of performing daily physical activity (OR: 0.4, *p* = 0.00) and limiting salt consumption (OR: 0.6, *p* = 0.04) are both negatively associated with BC. Regarding oral contraceptive use, no associations were found.

### 3.3. Social Network Analysis

We studied a data-driven approach by applying CD algorithms to identify relationships among risk factors potentially associated with the likelihood of having BC cancer, in order to pick out specific risk profiles to be used for tailoring health education and prevention programs. We reanalyzed data within each profile performing CD analysis. The resulting communities via the Gephi implementation for the three profiles of the postmenopausal women are described below in terms of characteristics’ association and women’s numerosity. The number of nodes always includes women’s and characteristics’ nodes.

Non-modifiable factors (Profile 1)

With the Gephi CD implementation, the best results were obtained for four communities with Modularity equal to 0.159. The graph is composed of 5621 nodes: 5601 nodes represent women and 20 nodes represent characteristics. The graph layout and the communities’ composition are shown in Figure 2. The characteristics of the nodes are defined in Appendix A.

The smaller community is the blue one, including 48 women who mainly share the characteristic of having or have had both female and male relatives with BC (MaFeFamhist). This community of women isolates itself without the attribution of other specific risk factors included in the first profile.

The largest community is fuchsia, which relates 2615 women, sharing older age (over than 60 years), BI-RADS A and B breast density (BI-A and BI-B), the lower breast densities, no familiarity for BC (NoFamhist), longer reproductive period length (over 39 years: Fert39–40 and Fert41+), and more years from menopause onset (over 6 years: Menop6+).

In terms of numerosity, after the fuchsia, there is the yellow community (2312 women), which relates women in the previous decade of age (50–59 years) with BI-RADS C and D (the higher breast densities), familiarity with BC (male relatives: MaFamhist or female relatives: FeFamhist) and fewer years from menopause onset (less than 5 years: MenopUpTo5). The green community, including 626 women, connects the youngest age (40–49 years) with the lowest reproductive period length (15–35 years: Fert15–35).

Personal history features (Profile 2)

The best results were obtained for four communities. The graph is composed of 5637 nodes: 5601 nodes represent women and 35 nodes represent characteristics. The modularity index is 0.08. Compared to the first profile (5621 nodes, of which 20 nodes represent characteristics) the higher number of characteristics included in this profile generates more connections between nodes in different communities, therefore decreasing the modularity index value, which is still included in the range for unweighted and undirected graphs. The graph layout and the communities’ composition are shown in Figure 3. The characteristics of the nodes are defined in Appendix A.

The largest community is the yellow one (2255 women) that relates women, most of whom share the following characteristics: older age (more than 60 years), weight gain in menopause (WGainMNP_Yes), hormone replacement therapy use (TOS_Yes), pregnancies (Pregn_Yes), and breastfeeding (BrFeed_Yes), no use of oral contraceptives (OrContr_NO), underwent CRT (CRT_Yes), having 2 or more comorbidities (Comorb_2+), being overweight or obese (BMI_OverW, BMI_Obese), being unemployed or retired (Oc_Unempl, Oc_Retir), and having low school level (primary or secondary school level: Edu_PriSec).

In terms of numerosity, after the yellow one, there is the green community (2009 women). The green community includes women who mostly share the following characteristics: younger age (from 40 up to 54 years), no pregnancy (Pregn_NO) and no breastfeeding (BrFeed_NO), performed ovarian stimulation (OvStim_Yes), oral contraceptives use for over 6 years (OrContr_6+), had no CRT (CRT_NO), had 1 comorbidity only (Comorb_1), normal weight (BMI_NormW), highest education level (University or post graduate level: Edu_Univ), and are employed (Oc_Empl) or occupied with night shifts (Oc_NightSh).

The fuchsia community is composed of 1003 women who mostly share the following characteristics: intermediate age from 55 up to 59 years, no performance of ovarian stimulation (OvStim_NO), use of oral contraceptives up to 5 years (OrContr_UpTo5), not performing hormone replacement therapy (TOS_NO), not presenting any comorbidities (comorb_NO), and a high school education level (Edu_HigSch).

The smallest community is the blue one, which is composed of 334 women who mostly share the following characteristics: no gain weight during menopause (WGainMNP_No) and a BMI indicating underweight (BMI_UnderW). In particular, underweight concerns a very low percentage of women within the overall network.

Lifestyle habits (Profile 3)

The best results were obtained for five communities with a Modularity of 0.137. The graph is composed of 5633 nodes: 5601 nodes represent women and 32 nodes represent characteristics. The graph layout and the communities’ composition are shown in Figure 4. The characteristics of the nodes are defined in Appendix A.

The largest community is the blue one (1852 women) that relates women most of whom share the following characteristics: middle age from 50 to 59 years, being low alcohol drinkers (Drink_Low), being strong smokers (Smok_Hig) but highly adherent to WCRF’s recommendations (high quantity of vegetables, cereals, fruit, legumes in diet: VegFood_Hig; very varied diet: VaryDiet_Hig; very limited use of salt in diet: LimSalt_hig; diet with very limited energy-dense foods and sugary drinks: LimCalFood_High; very limited use of red/cured meat: LimRedMeat_Hig), high levels of physical activity (PhisAct_High).

In terms of number of women included in each community, after the blue one there is the yellow community (1699 women) which relates women who mostly share the following characteristics: younger age from 40 to 49 years, medium alcohol drinkers (Drink_Med), adherent to WCRF’s recommendations (medium quantity of vegetables, cereals, fruit, legumes in diet: VegFood_Med; varied diet: VaryDiet_Med; limited use of salt in diet: LimSalt_Med; diet with limited energy-dense foods and sugary drinks: LimCalFood_Med; limited use of red/cured meat: LimRedMeat_Med), medium levels of physical activity (PhisAct_Med), and low level smokers (Smok_Low).

The azure community is composed of 1125 women who mostly share the following characteristics: heavy alcohol drinkers (Drink_High), very limited adherence to WCRF’s recommendations (low quantity of vegetables, cereals, fruit, legumes in diet: VegFood_Low; low varied diet: VaryDiet_Low; low limitation of salt in diet: LimSalt_Low; diet with low limited energy-dense foods and sugary drinks: LimCalFood_Low; low limitation in use of red/cured meat: LimRedMeat_Low), and low level of physical activity.

The fuchsia community is composed of 555 women who mostly share the following characteristics: age over 65 years, so the oldest women that are non-smokers (Smok_NO) and do not drink alcohol (Drink_NO).

The smallest community is the green one, composed of 370 women mostly are aged from 60 to 64 years and are moderate smokers (Smok_Med).

### 3.4. Combination of Profiles

To join the results coming from the previous steps in a comprehensive vision which includes all the profile evaluations, we characterize each woman based on her belonging to the three profiles simultaneously. We obtained 73 different combinations where the frequency distribution of women ranged from 0.02% to 6.6% (368 women; 1 BC case) and the BC prevalence distribution from 0.0% to 28.6% (7 women; 2 BC cases).

In order to increase the statistical power of our analysis, we focused on those groups of communities across all the profiles that share non-modifiable (profile 1) and personal history characteristics (profile 2) but differ in terms of lifestyle habits (profile 3). These groups are indicated as families.

Three different families were statistically associated with the BC rate (Table 1, Table 2 and Table 3; chi-squared test *p* = 0.04): the Fuchsia, Green, and Yellow families.

The Fuchsia family included 1299 women who share both the fuchsia community within profile 1 (non-modifiable factors) and the yellow community within profile 2 (personal history features). They accounted for 38 BC cases (prevalence rate: 2.5% [95%CI: 1.7–3.5%]; adj. residual = 2.1).

The Green family included 166 women who share both the green community within profile 1 and the yellow community within profile 2. They accounted for 7 BC cases (prevalence rate: 4.2% [2.0–8.4%]; adj. residual = 2.4).

The Fuchsia showed the highest percentage of women older than 60 years of age (79.1%), with lower breast density (BI-RADS A: 28.6%; and BI-RADS B = 45.7%), and the highest percentage of women without familiarity with BC (78.6%, adj. residual > 2), as shown in Table 1. Observed BC cases exceeded the expected (Table 1 and Figure 5) within the oldest women (chi-square test *p* = 0.013; adj. residual = 3.2).

The Green family (Table 1) included slightly fewer elderly women (average age 61.3 ±9.4 years) compared to the Fuchsia ones, with a high percentage of BI-RADS B density (45.8%). Within the Green family both age and breast density were not associated with BC cases (Table 1 and Figure 5).

The Fuchsia and the Green families shared some characteristics. Women were less qualified (primary/secondary education: 34.4% and 36.1%, respectively), unemployed or retired (63.8% and 57.2%, respectively), less healthy (2 or more comorbidities: 68.0% and 66.9%), with both families’ women particularly associated with neuropsychiatric disease (44.5% and 53.6%, respectively), and the Fuchsia women statistically associated to cardiometabolic (70.4%), intestinal (13.0%), and autoimmune (30.1%) diseases as well. Among the Green women, the employed and with neuropsychiatric diseases were both negatively associated with BC (Table 2 and Figure 5).

The Fuchsia and the Green families also showed the highest percentages of women who had pregnancies (88.8% and 86.1%, respectively) and experienced breastfeeding (73.2% and 69.3%, respectively); pregnancy was positively associated with BC within the Fuchsia family (chi-square test *p* = 0.042; adj. residual = 0.042) and negatively associated (chi-square test *p* = 0.023; adj. residual = −2.3) within the Green one (Table 2 and Figure 5).

Additionally, the Fuchsia and the Green families shared the positive association with some known risk factors for BC as a past experience with CRT (13.7% and 18.1%, respectively), higher body mass index (overweight: 29.6% and 28.9%, obesity: 13.5% and 12.0%, weight gain during menopause: 67.9% and 70.5%), and hormone replacement therapy use (more than six months 23.9% and 38.6%, respectively). Concerning the use of hormones for oral contraception, women belonging to the Fuchsia and the Green families were positively associated with the low level of exposure (never or less than 1 year of use: 59.3% and 59.0%, respectively).

Among the lifestyle habits, the Fuchsia and the Green families were both positively associated with a scarce adherence to the WCRF’S recommendation of limiting energy dense food, but slightly differed in terms of smoking habits, as the Fuchsia women showed a higher frequency of use (more than 5 cigarettes/day 29.0% vs. 27.7% of the Green family) and, in particular, the positive association with the frequency of 6–10 cigarettes/day (14.7%, adj. residual = 2.6). Furthermore, the Fuchsia women declared a worse compliance with the WCRF’s suggestion of staying physically active (negative association and lower percentage of women declaring the highest level of compliance) and the recommendation of limiting salt consumption (observed number of women lower than expected in the low level of compliance). Otherwise, the Fuchsia women showed a better compliance with the suggestion of following a varied diet (medium and high compliance 93.5% vs. 89.2% of the Green family), even though they showed negative association with the highest level of compliance.

The Yellow family had the lowest BC rate (prevalence rate: 0.9% [95%CI: 0.5–1.8%]; adj. residual = −2.2) significantly different from the other families (Table 1). It included 889 women who shared both the yellow community within profile 1 and the green community within profile 2 and accounted for 8 BC cases.

The Yellows were younger than the Fuchsias and the Greens (the number of observed women in the Yellow family aged 50–59 years was statically higher than expected), but ageing 60–64 years was positively associated with BC (Table 1 and Figure 5). The Yellow women showed a higher breast density (BI-RADS C: 46.5% and BI-RADS D: 18.6%) and a positive association with BC familiarity, in particular with first line females (29.7%, adj. residual = 3.4) and male (0.9%; ad. Residual = 2.2) relatives with BC (Table 1). As shown in Table 1 and in Figure 5, having first line male familiarity was positively associated with BC (chi-square test *p* = 0.002; adj. residual = 3.5) as well as being in the menopause status for more than five years (chi-square test *p* = 0.031; adj. residual = 2.2).

The Yellow women were the most qualified (university or postgraduate degree 41.3%, adj. residual = 4.5), employed (73.1%, adj. residual = 9.8), with the highest rate of night shifts workers (7.2%; adj. residual = 9.9), and with the lowest rate of women who experienced pregnancies (68.4%) and breast feeding (53.5%). Women who experienced breast feeding (Table 2 and Figure 5) showed a negative association with BC (chi-square test *p*= 0.028; adj. residual = −2.3). For the large majority (80.9%), the Yellows showed a normal weight and are negatively associated with the most extreme BMI classes (overweight and obese) and the weight gain during menopause (54.2%, adj. residual = −2.7). More than half of the Yellow women showed more than 2 comorbidities (lower rate than the expected one and the lowest rate compared to all the other women) with a negative association with cardiometabolic (51.5%, adj. residual = −4.7), intestinal (8.8%; adj. residual = −2.6), and autoimmune diseases (21.8%; adj. residual = −2.1). They showed a more frequent use of hormones for ovarian stimulation (14.1%, adj. residual = 9.1) and a longer use of oral contraceptives (more than 5 years: 36.2%, adj. residual = 6.8%), but the lowest frequency of hormone used as replacement therapy (more than 6 months: 14.6%; adj. residual = −3.9).

Compared to the other women, the Yellow ones seemed to be more focused on lifestyle habits. Generally, they showed the highest levels of compliance to the WCRF’s recommendations, in particular with the suggestion of staying physically active (high level of compliance: 31.9%, adj. residual = 2.5). They also showed healthier habits concerning alcohol drinking.

To better identify the association between lifestyle habits and BC rate, we also compared the five combinations of profiles belonging to the Fuchsia family (Appendix A). No statistically significant difference was detected among the BC rates of the five combinations (chi-square test *p* > 0.05). The Fuchsia-Yellow-Fuchsia (FYF) combination was the only one showing a BC rate (4.7% [2.1–8.7%]: 9 BC cases among 193 women) significantly higher than the BC rate of the other combinations’ group belonging to the Fuchsia family (Appendix A: chi-square test *p* = 0.033).

Within the Fuchsia family, FYF women were 5 years older than the other women (Appendix A: 68.5 ± 6.5 vs. 63.8 ± 6.8) with less dense breast tissues (BI-RADS A and B), as well as the other Fuchsia women. They had a comparable reproductive period length (38.5 ± 3.8 vs. 38.9 ± 3.9), but they were in menopause for a longer time (17.5 ±7.8 years vs. 12.6 ± 7.4 among the other Fuchsia; chi-square test *p* < 0.001). Furthermore, they showed the highest rate of comorbidity (Appendix A: 71.0% vs. 67.5%; chi-square test *p* > 0.05) with a statistically significant association concerning autoimmune diseases (Appendix A: 36.3% vs. 29.0% among the other Fuchsia, chi-square test *p* = 0.043). The FYF rate of overweight and obesity is slightly higher (Appendix A: 46.1% vs. 42.6%) even though less FYF women declared to have experienced weight gain during menopause (Appendix A: 63.2% vs. 68.7%; chi-square test *p* > 0.05).

Furthermore, the FYFs were less qualified (primary, secondary education, adj. residual > 2), unemployed (Appendix A: adj. residual > 2), never used oral contraception or used it for less than one year (Appendix A: chi-square test *p* < 0.001), and were positively associated with high compliance to the WCRF’s recommendations on red meat, plant food diet, and salt (Appendix A). Additionally, they were negatively associated with a low compliance with the WCRF’s suggestion of following a varied diet. Their habits towards tobacco use and alcohol consumption are both protective (highest rates of never smokers and no alcohol drinkers).

The FYF’s BC cases were positively associated with low compliance with daily physical activity both when comparing FYF to all the other Fuchsia combinations and to the group of all the other Fuchsia women (Appendix A and Figure 5).

Another interesting combination belonging to the Fuchsia family was the Fuchsia-Yellow-Blue one (FYB), the most populous, whose features are shown in Appendix A. Notwithstanding that the FYB’s BC rate did not differ from that of all the others Fuchsia women (Appendix A: 2.3% [1.0–4.6%]), in the comparison with the other Fuchsia combinations, it was positively associated with some specific history features (Appendix A and Figure 5) as being retired (chi-square test *p* = 0.030; adj. residual = 2.4) and underweight (chi-square test *p* = 0.011; adj. residual = 3.0). Additionally, the FYB’s BC rate was positively associated (Appendix A and Figure 5) with low compliance to the suggestion of limiting the consumption of red meats and cured meats (chi-square test *p* = 0.020; adj. residual of low compliance = 2.2).

The FYB women, on average, are 5 years younger than the FYF ones, more qualified and employed (Appendix A: both chi square test *p* < 0.001), and with the highest rates (Appendix A: chi-square test *p* = 0.011) of normal weight (64.3%) and underweight (1.2%). They declared the highest compliance with all the WCRF’s recommendations, but they also showed the highest quote of more frequent smokers (Appendix A: more than 10 cigarettes/day: 21.9%), different from that of the other families (chi-square test *p* < 0.001).

## 4. Discussion

It is well known and evidence-proved that behavioral education and information can strongly influence health patterns; in fact, the highest levels of health literacy in this field are associated with better health outcomes [47]. The relationship between health literacy, health promotion, and public health became even more crucial during the COVID-19 pandemic, which brought out the critical issues of public health messages aimed to raise population attention on specific attitudes and preventive behaviors [48]. However, designing effective risk reduction interventions is the final step of a process that starts with disease definition, passes through the detection of negative/predisposing behaviors, and finally lands to the formulation of possible solutions. This requires knowledge of previous research on the outcomes, but also of the key mechanisms of perception, knowledge, risk, and attitudes which influence behaviors [49].

Evidence from the literature shows that healthy lifestyle behaviors tend to cluster. People following a healthy diet often have a healthy weight, exercise regularly, have a moderate alcohol intake, and do not smoke. [24]. On the other hand, the same literature stated that defining the true cancer protective effects of lifestyle choices without confounding factors can only be achieved through large (over 25,000 patients) randomized controlled trials (RCT), which are often prohibitively expensive. Furthermore, BC does not have the well-defined surrogate endpoints as it happens with cholesterol and blood pressure for evaluating whether lifestyle interventions can prevent CVD. This makes cancer prevention guidelines more likely identifiable through large cohort studies rather than through RCT [24].

Our study is in line with these statements, as we aimed to identify combinations of non-modifiable and lifestyle related factors, which could influence the chance of having BC in postmenopausal women, in order to pick out some risk profiles to be used for tailoring health education and prevention programs. In particular, we used a structured self-administered questionnaire investigating risk factors, family history, and lifestyle to collect women’s personal information to be linked with clinical data (histopathological proven BC and their features) provided by clinicians.

The data analyses followed a twofold strategy. First, within the overall population of postmenopausal women (5601 women), we assessed the weight of each risk factor on the likelihood of developing BC using a binomial logistic model. As a second step, we identified relationships among risk factors potentially associated with the likelihood of having BC cancer in order to profile communities of women who share more dense connections among some specific features compared to women not belonging to these communities (SNA). Then, once the communities were identified for each profile, we characterized each woman based on the combination of communities she belongs to along the three profiles. To the aim of investigating the role of the lifestyle habits in BC onset, we grouped the 73 combinations into 15 families that share non-modifiable (profile 1) and personal history features (profile 2). We focused on the families showing a statistically significant association with the BC rate (Fuchsia, Green and Yellow) by comparing their risk factors’ distributions through the chi-square test and the adjusted standardized residual analysis. Finally, we analyzed more in depth the combinations of profiles belonging to the Fuchsia family (FYF and FYB), whose BC rate had a significant association with some risk factors.

The most relevant result in terms of lifestyle habits influencing the risk of BC concerns the WCRF’s recommendation of staying physically active every day (Figure 5). Both the binomial logistic model, applied to the overall menopause women’s network (Table 4: OR: 0.4, *p* < 0.001), and the SNA show that a better compliance to this recommendation is associated with a lower risk of BC onset, particularly among women belonging to other families than the Fuchsia, Green, and Yellow (Table 3). It is also worth noting that the FYF women who declared a low compliance with the recommendation (Appendix A and Figure 5) are positively associated with BC. This result is particularly relevant because the FYF combination shows the highest BC rate compared to the group of all the other Fuchsia combinations (4.7%; 95%CI [2.1–8.7%] vs. 2.1 [1.7–3.5%] among the other Fuchsia women; chi-square test *p* = 0.033) and also in light of the FYF women’s features (Appendix A) as older age, longer menopause, lower education, highest rate of unemployed, association with autoimmune diseases, null or very short use of oral contraception, healthy smoking and alcohol drinking habits, and being associated to a generally high compliance with all the WCRF’s recommendations except for that on physical activity. Conversely, the FYF women do not statistically differ in terms of BMI or weight gain during menopause, if compared to the group of all the other Fuchsia combinations taken as a whole (Appendix A).

Another relevant result concerns the FYB combination within the Fuchsia family (Appendix A). The FYB women are slightly younger than the FYF women (Appendix A, FYF: 64.5 (±6.9) years; FYB: 63.2 (±7.0) years). They have the highest propension towards tobacco use (Appendix A: more than 6 cigarettes/day 21.9%), also compared to women not belonging to the Fuchsia family (Table 3: 12.8%). Additionally, compared to the other Fuchsia combinations, FYB women have the highest rate of underweight (1.2%) and declared the highest compliance with all the WCRF’s recommendations (more than 71.0% for all the WCRFs, except for physical activity: Appendix A). Among the FYB women, exceeding BC cases are associated with being underweight and retired (Appendix A and Figure 5). Additionally, exceeding BC cases are associated with the few women having a scarce adherence to the suggestion of limiting red meat and cured meat consumptions (Appendix A and Figure 5).

The BC rate of women not belonging to the Fuchsia, Green, and Yellow families is positively associated with the oldest age (Table 1 and Figure 5), null or low exposure to hormones for oral contraception and with past experience of chest radiation therapy, even though the CRT exposure concerned the 6.5% of them only (Table 2 and Figure 5). The highest adherence to the suggestion of limiting salt consumption was negatively associated with BC (Table 3 and Figure 5), according to the risk reduction of 38% detected through the binomial regression model (Table 4).

Our results from the logistic model (overall network of women) and the SNA (women not belonging to the Fuchsia, Green, and Yellow families and FYF women) agree with previous systematic review meta-analyses, which indicate physical activity as a strong protective factor against BC risk [50,51]. Pizot C et al. [52] found that HRT use hampers the preventative effect of physical activity. In our study this finding could partly explain the high BC prevalence rate among the Green family woman, showing both the highest rate of HRT use (38.6%, Table 2) and the highest rate of low-compliance with physical activity (41.6%), as reported in Table 3.

Findings about the FYB women were also in line with the International Agency for Research on Cancer indications, reporting that red meat and cured meat may be potential carcinogens for humans [53]. A recent meta-analysis estimated a 10% BC risk increase associated with the consumption of red meat, 7% for total meat, and 18% for cured meat [54]. The carcinogenicity of red meat and processed meat may be attributed to mutagenic compounds, such as polyaromatic hydrocarbons and heterocyclic amines, which are by-products of cooking red meat at high temperatures [55]. Furthermore, heme iron, fat, and animal sugar molecule N-glycolylneuraminic acid found in red meat, as well as hormone residues due to the stimulation of more rapid growth in beef induce increased inflammation and oxidative stress, which have been suggested as independent risk factors of BC [55]. Considering that increased subcutaneous fat and energy reserves may help to overcome the catabolic changes due to neurohormonal and inflammatory pathways, the carcinogenic effect could be particularly relevant among underweight women, also identified as at greater risk of overall infection, infection of the respiratory tract and the skin, and multi-organ dysfunction [56].

Finally, concerning women not belonging to the Fuchsia, Green, and Yellow families, our findings agree with those from a population-based cohort study conducted in France, focusing on the association between ultra-processed food intake and a higher overall cancer and specific BC risk [57]. Among other compounds, ultra-processed foods often have a higher content of added salt along with a lower fibre and vitamin density, and their consumption is often associated with a higher risk of hypertension. Additionally, a more recent in vivo (MMTV-PyVT mice) study showed that a salt-rich diet accelerates the progression of BC by increasing the proportion of Th17 lymphocytes. Increased Th17 lymphocytes concentration, via the secretion of IL-17F cytokine, which activates the MAPK signaling pathway in BC cells, leads to the unregulated expression of the pro-tumor genes, which finally accelerates tumor progression [58].

Although our results look promising, we outline some limitations in the present study. The major limitation is the current low number of BC cases (100) and the scarce availability of the questionnaires, which have highly impacted the statistical analysis power. The recent pandemic has resulted in delays, both in recruiting women/collecting clinical data and in returning the filled questionnaires. Additionally, the generalisability of our results to the general population is limited by the self-selection bias of the P.I.N.K. women who are more qualified and employed than the Italian women as a whole. This suggests that they could be more prone to protective habits and could have more resources to be dedicated to their health. The third limitation concerns the use of a paper questionnaire for self-reporting data, which could have impacted the data availability and the accuracy and completeness of our results. These limitations suggest that the current findings may be considered preliminary even though we found them very relevant and essential to illustrate an alternative method of analyses (the data driven approach) and for the development of an innovative tool, the Dress-PINK, designed to collect population-based information in an innovative way. In the next few months, the Dress-PINK could potentially address the entire population of the Italian adult women and allows us to collect complete individual data about eating (semi-quantitative data about dietary patterns, food quality choice, and cooking habits) and other lifestyle habits, based on widely validated scales (Medi-Lite, Dass-21, Lifetime Physical Activity Questionnaire, International Physical Activity Questionnaire, Short-Form Health Survey-12, etc.). The Dress-PINK is a customization of the Dress system (“Doing Risk sElf-assessment and Social health Support”), a mobile-health system based on the Telegram bot and developed during the recent COVID-19 pandemic [59]. The Dress system is an innovative and cost-effective approach to collect health data and run risk assessment on users’ smartphones via Telegram Messenger. Its basic idea is to establish a lasting link between the user and the tool, thus enabling the modeling of the data to assess the individual risk of various health conditions and to improve the individual’s self-empowerment. Therefore, the tool asks users a set of questions grouped into daily clusters, formulated by reviewing the most relevant scientific references. Furthermore, tailored prevention messages are provided to promote critical consciousness, critical thinking, and increased health literacy [60]. This is an essential step for a progressive change from an early diagnosis only-based approach to a personalized preventive and risk-reducing one.

## 5. Conclusions

Defining the true anticancer effects of lifestyle choices is particularly hard because it needs a large amount of data to overcome the high variability and the complex interactions that characterized the general population. The P.I.N.K. framework represents an example of integrating and analyzing in a new way a huge amount of clinical data together with data on lifestyle and daily habits to emphasize their crucial role in the development of BC. It also shows the importance of defining the effects of preventive/predisposing behavior combinations to identify targeted risk profiles. In our opinion, the collaboration between stakeholders and researchers for building real-world evidence and the scientific-based development of mobile health tools could be the winning strategies for promoting tailored prevention policy and health education programs to improve communities’ self-empowerment.

## Figures and Tables

**Figure 1 cancers-14-05801-f001:**
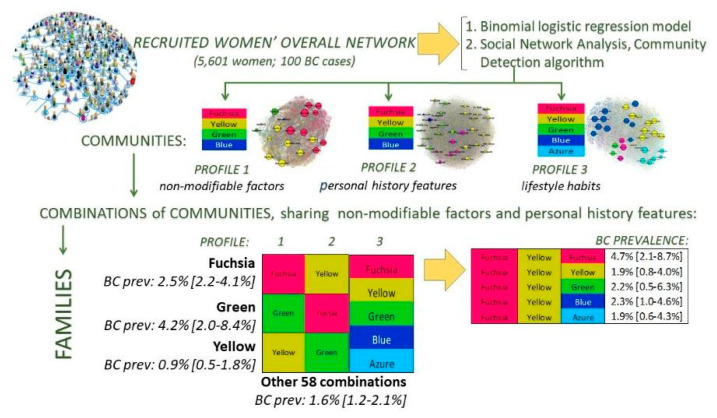
Overview of the statistical analysis approaches.

**Figure 2 cancers-14-05801-f002:**
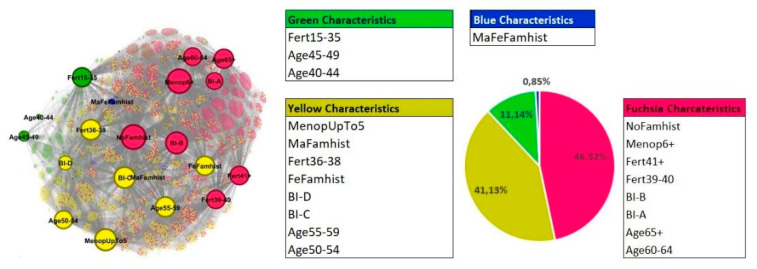
Graph layout and composition of communities within the profile 1. The pie chart represents the percentages of women belonging to each community.

**Figure 3 cancers-14-05801-f003:**
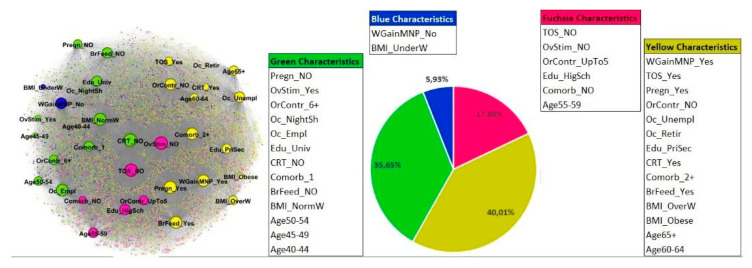
Graph layout and composition of communities within the profile 2. The pie chart represents the percentages of women belonging to each community.

**Figure 4 cancers-14-05801-f004:**
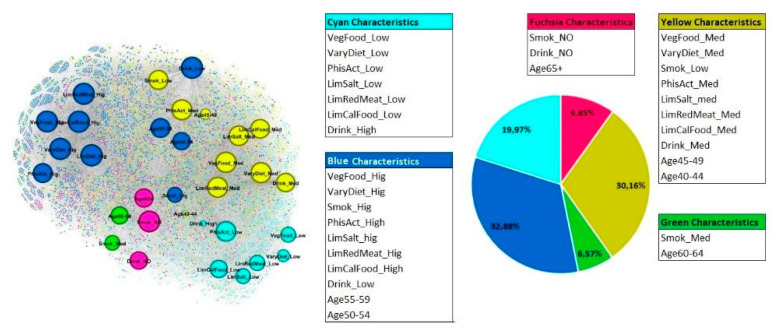
Graph layout and composition of communities within the profile 3. The pie chart represents the percentages of women belonging to each community.

**Figure 5 cancers-14-05801-f005:**
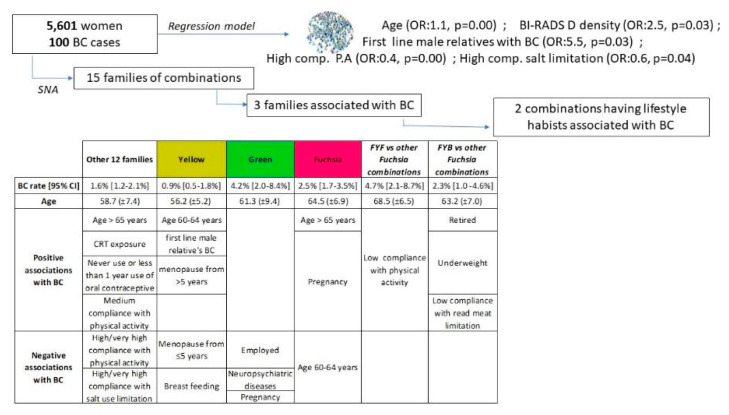
Graphical overview of the main results.

**Table 1 cancers-14-05801-t001:** Distribution of the characteristics included in the profile 1: overall population of women and focus on the families of community combinations statistically associated with BC. See Section 3.4 for the description of the communities and the families.

	Women Belonging to Other Families	Fuchsia Family	Yellow Family	Green Family	Chi-Square Test	All Women
Number of women	3247	1299	889	166		5601
BC prevalence *[95% CI]*	1.6% [1.2–2.1%]	2.5% [1.7–3.5%] (b)	0.9% [0.5–1.8%] (a)	4.2% [2.0–8.4%] (b)	*p* = 0.004	1.8% [1.5–2.2%]
Years of age (avg ± SD) *focus on age classes*	58.7 (±7.4); > 65 * (b)	64.5 (±6.9); 60–64 ** (a); >65 ** (b)	56.2 (±5.2); 50–54 **; 55–59 **; 60–64 * (b)	61.3 (±9.4); 45–49 **; 55–59 *; >65 **	*p* < 0.001	59.7 (±7.6)
Breast density:					*p* < 0.001	
BI-RADS A (less dense)	14.6% *	28.6% **	7.3% *	21.7%		16.9%
BI-RADS B	37.1%	45.7% **	27.7% *	45.8% **		37.9%
BI-RADS C	36.8% **	23.9% *	46.5% **	25.9% *		35.0%
BI-RADS D	11.5% **	1.7% *	18.6% **	6.6%		10.2%
Family history of BC:					*p* < 0.001	
no first line relatives	72.5% *	78.6% **	69.4% *	76.5%		73.5%
first line female relatives	25.5%	21.4% *	29.7% **	23.5%		25.2%
first line male relatives	0.5%	0.0% *	0.9% ** (b)	0.0%		0.4%
first line male and female rel.	1.5%	0.0%	0.0%	0.0%		0.9%
Menarche age (avg. years ± SD)	12.6 (±1.4)	12.4 (±1.4)	12.7 (±1.4)	12.8 (±1.5)		12.6 (±1.4)
Reproductive period length	37.4 (±4.2)	38.8 (±3.9)	37.7 (±3.3)	32.1 (±3.4)		37.6 (±4.1)
Menopause (avg. years ± SD). Focus on length classes	8.6 (±7.3) ≤5 **; >5 *	13.3 (±7.7) ≤5 *; >5 **	5.9 (±5.7); ≤5 ** (a); >5 * (b)	16.5 (±10.2) ≤5 *; >5 **	*p* < 0.001	9.5 (±7.7)

Association between each risk factor and family (chi-square test *p* < 0.05): * negative (adj. residual < −2); ** positive (adj. residual > 2); Association between BC cases and each risk factor by family (chi-square test *p* < 0.05): (a) negative (adj. residual < −2); (b) positive (adj. residual > 2).

**Table 2 cancers-14-05801-t002:** Distribution of the characteristics included in the profile 2: overall population of women and focus on the families statistically associated with BC. See Section 3.4 for the description of the communities and the families.

	Women Belonging to Other Families	Fuchsia Family	Yellow Family	Green Family	Chi-Square Test	All Women
Number of women	3247	1299	889	166		5601
BC prevalence [95% CI]	1.6% [1.2–2.1%]	2.5% [1.7–3.5%] (b)	0.9% [0.5–1.8%] (a)	4.2% [2.0–8.4%] (b)	*p* = 0.004	1.8% [1.5–2.2%]
Qualification:					*p* < 0.0010	
primary secondary education	24.1% *	34.4% **	19.1% *	36.1% **		26.1%
high school graduation	40.9% **	35.0% *	39.6%	34.9%		39.2%
university/post graduate	35.0%	30.6% *	41.3% **	28.9%		34.8%
Occupation:					*p* < 0.001	
unemployed/housewife	33.3% *	59.3% **	19.7% *	53.6% **		37.8%
employed	63.8% **	36.1% *	73.1% **	42.8% * (a)		58.2%
retired	0.6% *	4.5% **	0.0% *	3.6% **		1.5%
work/worked night shifts	2.3%	0.0% *	7.2% **	0.0% *		2.5%
Current or past Co-morbidities:					*p* < 0.001	
1 co-morbidity	27.8%	24.9% *	35.5% **	26.5%		28.3%
at least 2 co-morbidities groups	56.3% *	68.0% **	51.9% *	66.9% **		58.6%
Types of comorbidities:						
cardiometabolic diseases	55.6% *	70.4% **	51.5% *	63.3%	*p* < 0.001	58.6%
endocrine diseases	49.5%	52.1%	53.1%	56.0%	*p* > 0.05	50.9%
intestinal diseases	11.3%	13.0% **	8.8% *	11.4%	*p* = 0.024	11.3%
neuropsychiatric diseases	39.3% *	44.5% **	40.7%	53.6% ** (a)	*p* < 0.001	41.2%
autoimmune diseases	23.3% *	30.1% **	21.8% *	21.7%	*p* < 0.001	24.6%
Chest radiation therapy	6.5%* (b)	13.7% **	4.5% *	18.1% **	*p* < 0.001	8.2%
Pregnancy (at least one)	80.0%	88.8% ** (b)	68.4% *	86.1% (a)	*p* < 0.001	80.4%
Breast feeding	63.1%	73.2% **	53.5% * (a)	69.3%		64.1%
BMI:					*p* < 0.001	
underweight (<18.5)	6.1% **	0.4% *	0.3% *	0.0% *		3.7%
normal weight (18.5–24.9)	68.2%	56.5% *	80.9% **	59.0% *		67.2%
overweight (25–29.9)	20.5% *	29.6% **	15.0% *	28.9% **		22.0%
obese (≥30)	5.2% *	13.5% **	3.8% *	12.0% **		7.1%
Weight gain in menopause	55.0% *	67.9% **	54.2% *	70.5% **	*p* < 0.001	58.3%
Oral contraceptive use:					*p* < 0.001	
never or less than 1 year	46.9% * (b)	59.3% **	42.4% *	59.0% **		49.4%
1–5 years	25.1% **	22.3%	21.4%	18.7%		23.7%
more than 5 years	28.0% **	18.4% *	36.2% **	22.3%		26.9%
Ovarian stimulation:					*p* < 0.001	
at least 1 cycle of stimulation	6.8%	2.9% *	14.1% **	3.0%		6.9%
HRT:					*p* < 0.001	
more than 6 months	17.9% *	23.9% **	14.6% *	38.6% **		19.4%

Association between each risk factor and family (chi-square test *p* < 0.05): * negative (adj. residual < −2); ** positive (adj. residual > 2); Association between BC cases and each risk factor by family (chi-square test *p* < 0.05): (a) negative (adj. residual < −2); (b) positive (adj. residual > 2).

**Table 3 cancers-14-05801-t003:** Distribution of the characteristics included in the profile 3: overall population of women and focus on the families statistically associated with BC. See Section 3.4 for the description of the communities and the families.

	Women Belonging to Other Families	Fuchsia Family	Yellow Family	Green Family	Chi-Square Test	All Women
Number of women	3247	1299	889	166		5601
BC prevalence [95% CI]	1.6% [1.2–2.1%]	2.5% [1.7–3.5%] (b)	0.9% [0.5–1.8%] (a)	4.2% [2.0–8.4%] (b)	*p* = 0.004	1.8% [1.5–2.2%]
Smoking habits:					*p* = 0.010	
never-smokers	42.7%	42.7%	46.5%	47.6%		43.4%
1–5 cigarettes/day	32.2% **	28.3% *	29.2%	24.7%		30.6%
6–10 cigarettes/day	12.4%	14.7% **	10.9%	10.2%		12.6%
more than 10 cigarettes/day	12.8%	14.3%	13.4%	17.5%		13.4%
Alcohol drinking habits:					*p* = 0.005	
no alcohol drinker	20.5%	24.1% **	19.7%	25.9%		21.4%
one time per week or less	53.0%	47.7% *	56.6% **	45.8%		52.1%
more times per week	26.2%	27.9%	23.5%	28.3%		26.2%
binge-drinking at least one time/week	0.2%	0.4%	0.2%	0,0%		0.3%
WCRFs compliance						
1. Stay daily physically active:					*p* < 0.001	
low	33.2%	35.6%	30.8%	41.6% **		33.6%
medium	37.4% (b)	40.5% **	37.2%	28.3% *		37.8%
high/very high	29.4% (a)	23.9% *	31.9% **	30.1%		28.5%
2. Limit energy-dense foods:					*p* > 0.05	
low	20.0% *	23.6% **	19.8%	33.7% **		21.2%
medium	28.6%	29.0%	27.0%	21.1% *		28.2%
high/very high	51.4%	47.4% *	53.2%	45.2%		50.6%
3. Follow a plant-based diet:					*p* > 0.05	
low	13.1%	13.2%	11.8%	19.3%		13.1%
medium	40.8%	42.2%	41.2%	38.0%		41.1%
high/very high	46.1%	44.6%	47.0%	42.8%		45.8%
4. Limit red/cured meats:					*p* > 0.05	
low	13.5%	14.9%	15.5%	14.5%		14.1%
medium	37.0%	39.9%	35.5%	41.6%		37.6%
high/very high	49.5%	45.3%	48.9%	44.0%		48.3%
5. Limit salt consumption:					*p* = 0.003	
low	11.8% *	14.7% **	13.7%	17.5%		12.9%
medium	34.5%	36.3%	30.9% *	32.5%		34.3%
high/very high	53.7% (a)	49.0%*	55.3%	50.0%		52.8%
6. Follow a varied diet:					*p* = 0.012	
low	7.7%	6.5%	6.1%	10.8%		7.2%
medium	39.1%	43.5% **	38.0%	41.0%		40.0%
high/very high	53.2%	50.0% *	55.9%	48.2%		52.8%

Association between each risk factor and family (chi-square test *p* < 0.05): * negative (adj. residual < −2); ** positive (adj. residual > 2); Association between BC cases and each risk factor by family (chi-square test *p* < 0.05): (a) negative (adj. residual < −2); (b) positive (adj. residual > 2).

**Table 4 cancers-14-05801-t004:** Multivariate Logistic Analysis: protective or predisposing factors effects on BC (all menopause women).

		OR (95% CI)	*p*
**Age**		1.08 (1.06–1.11)	0.00
**Brest density:**	BI-Rads A	reference	
	BI-Rads B	1.78 (0.97–3.29)	0.06
	BI-Rads C	1.43 (0.74–2.75)	0.29
	BI-Rads D	2.52 (1.10–5.77)	0.03
**Oral contraceptive:**	Never used	reference	
	1–5 years	0.64 (0.37–1.11)	0.11
	more than 5 years	0.56 (0.31–1.00)	0.05
**First line relatives with BC:**	none	reference	
	female	1.16 (0.75–1.81)	0.50
	male	5.55 (1.24–24.93)	0.03
	male and female	0.00 (0.00–0.00)	1.00
**WCRF physical activity:**	medium compliance	reference	
	low compliance	0.85 (0.55–1.32)	0.48
	high compliance	0.41 (0.22–0.76)	0.00
**WCRF salt:**	medium compliance	reference	
	low compliance	0.93 (0.52–1.66)	0.81
	high compliance	0.62 (0.40–0.97)	0.04

## Data Availability

The PINK study data sharing is not applicable as specified in the informed consent signed by women.

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
