# Peer review of "Promote Community Engagement in Participatory Research for Improving Breast Cancer Prevention: The P.I.N.K. Study Framework"

_cancers, 2022, doi:10.3390/cancers14235801_

Round 1

Reviewer 1 Report

This is a well written paper on an important topic such as the tailoring of breast cancer prevention by using  community engagement.

The approach used by the authors is very innovative and interesting based on a multistep analysis and social network tecniques , the results are scientifically soundings.

However it is mandatory that the authors add some points in the results section and to discuss a possible limitation to the generalisability of the results to the general population : the self selection bias of the women included in the cohort.

Women “presenting spontaneously for routine breast examination ...”   are not a random sample of the general population, usually belong to a “community” more used to healthy lifestyles. This bias may be amplified for the women that decided to respond to the study questionnaires.  Also the cancers found by this kind of opportunistic screening are not a random sample of all the cancers, usually small cancers without evident clinical manifestation. It may be interesting if the authors could add few rows using available data to let the authors to understand to which degree  the women participating to the questionnaires are different from the general population, for example analysing women occupation and qualification respect to the general population. In the discussion this possible bias may be described by the authors. I also suggest to add some simple information on the clinical characteristic of the BC cases found , for example stage.

Author Response

Dear Reviewer 1,

thank you for your precious comments. We have made substantial changes to the manuscript according to your indications and to the comments of the other two Reviewers. You will find them in the newly uploaded manuscript: the additions and deletions are in track changes mode. Bibliographic references are changed and improved, but we have already accepted the revisions to make the text more clear.

We completely changed the abstract, as suggested by the Reviewer 3. The updated abstract is a bit longer than the maximum number of words indicated in the authors guidelines.

We thoroughly upgraded the introduction section following the flow suggested by the Reviewer 3. We also better conceptualised the aims of the work, as suggested by the Reviewer 3 (Lines 305-310) and we added a paragraph about the study limitations in the discussion section (Lines 851-864).

Below we provide point by point responses to your questions:

  • However it is mandatory that the authors add some points in the results section and to discuss a possible limitation to the generalisability of the results to the general population: the self-selection bias of the women included in the cohort.

We are aware that the Pink women are not a random sample of the general population, but a selected cohort. It is worth noting that some of them attend the PINK centers  after having already performed a mammography within the population screening. In these cases, mammography is subjected to a second reading. In our opinion, this approach makes the PINK cohort more comparable to the general population. However, to compare women participating to the questionnaires and the general population we added some data from the Eurostat database. In particular we refer to the Lines 443-446 within the Results paragraph 3.1. Additionally,  we highlighted the selection bias limitation in the Discussion section (Lines 851-864).

  • I also suggest to add some simple information on the clinical characteristics of the BC cases found, for example stage.

As we state in the Material and Methods section (Line 360-363):

  • The output measure, the BC diagnoses provided by radiologists, has been defined as a dichotomous variable (yes/not) using the anatomic pathology reports of BI-RADS 5 (B5) lesions resulting from needle core biopsies.

so, the BC diagnosis is based on the anatomic pathology reports from needle core biopsies, where breast lesions are coded from B1 to B5, without any references to the stage of the disease. We also collect the final report of the histological test after surgery for breast cancer, but currently the 85% out these reports are available, only. Due to the low number of BC cases, we decided to work on the histological report from core biopsy. However, from a preliminary analysis about the concordance between the 418 overall core biopsy and surgery reports available about the same BC cases, we estimated the 97% of compliance in identifying malignant lesions (see Lines 363-366).

We believe all these changes have strongly improved our manuscript, as well as the bibliography upgrade and the English language revision.

Reviewer 2 Report

This very interesting manuscript describes preliminary results of a network analysis of the P.I.N.K. study, which is an ongoing longitudinal multi-center study in Italy. 

The authors used a twofold approach combining classical binomial logistic regression and the data driven network-analysis approach.

This manuscript confirms already published epidemiological data. Using a longer observational period breast cancer incidence might increase and can further underline the findings of the authors. Although only 100 breast cancer cases were detected, I recommend to accept the paper. However, minor revisions should be applied to the manuscript:

Please give a more detailed description of the excluded patients. 29355 patients were recruited overall, yet only 5601 patients were included into the analysis:

How many patients are premenopausal? How many postmenopausal?

Which postmenopausal patients did you exclude due to which rationale? Why did you only analyse patients that underwent more than one examination at a breast center?

Could this data selection possibly influence the results and interpretation of the data?

Why did you exclude premenopausal patients? I would be highly interested in a network analysis characterising this patient cohort! Especially, whether risk factors  of premenopausal vs. postmenopausal patients differ largely or if the same approaches should be applied to every patient population.

The tables state p=0.000; this is not scientifically accurate. If the p value is very small, I would suggest to use p < 0.001. This modification should be applied to all tables and in the text.

Please explain why you presented the descriptive statistics of the families prior to characterising them in detail. This is a little bit confusing for the reader. Changing the order of the presented results could facilitate the understanding of the manuscript.

Author Response

Dear Reviewer 2,

thank you for your precious comments. We have made substantial changes to the manuscript according to your indications and to the comments of the other two reviewers. You will find them in the newly uploaded manuscript: the additions and deletions are in track changes mode. Bibliographic references are changed and improved, but to make the text more clear we have already accepted the revisions.

We completely changed the abstract, as suggested by the Reviewer 3. The updated abstract is a bit longer than the maximum number of words indicated in the authors guidelines.

We thoroughly upgraded the introduction section following the flow suggested by the Reviewer 3. We also better conceptualised the aims of the work, as suggested by the Reviewer 3 (Lines 305-310) and we added a paragraph about the study limitations in the discussion section (Lines 851-864).

Below we provide point by point responses to your questions:

Please give a more detailed description of the excluded patients. 29355 patients were recruited overall, yet only 5601 patients were included into the analysis:

How many patients are premenopausal? How many postmenopausal?

Which postmenopausal patients did you exclude due to which rationale? Why did you only analyse patients that underwent more than one examination at a breast center?

Could this data selection possibly influence the results and interpretation of the data?

Why did you exclude premenopausal patients? I would be highly interested in a network analysis characterising this patient cohort! Especially, whether risk factors of premenopausal vs. postmenopausal patients differ largely or if the same approaches should be applied to every patient population.

To date, out of all the women who have presented spontaneously for routine breast examination at the sixteen P.I.N.K. diagnostic centers in Italy, 29355 women decided to participate in the study.

During the study:

  • radiologists collect clinical data through a web platform;
  • diagnostic centers collect the completed questionnaires about personal features and lifestyle habits and return them to the CNR, where the questionnaires are processed with optical reading technique.

Then, the two sources of data are linked by using the woman identification number.

In this work we only included women having available both the clinical and the self-reported data (10097 women: 5601 post-menopausal and 4446 premenopausal). Among the premenopausal women the BC cases were 57 only and this makes the application of statistical analyses too hard (Lines 353-356). We will perform the analysis among the premenopausal women once the BC number will be more consistent. We also plan to perform the same analysis about the postmenopausal women once the BC incidence will be increased.

We analysed women who underwent more than one examination because the study design is longitudinal (cohort study): we start from the exposure at the baseline (risk factors) and follow all the women to identify those who developed breast cancer and those who do not. We excluded women who underwent  a single examination at the PINK centers from 2018 to 2022 , because we consider them “lost at the F-Up”,  as they may have developed BC but we are unable to know their current health condition. Since within a longer observational period a woman currently defined as lost at the F-UP could attend a further evaluation in the same PINK center, we decided to estimate the selection bias effects at the end of the study, when the women lost at the F-UP will be definitively identified. 

The tables state p=0.000; this is not scientifically accurate. If the p value is very small, I would suggest to use p < 0.001. This modification should be applied to all tables and in the text.

Regarding the suggestion of changing the p value, we corrected all the Tables and the text.

Please explain why you presented the descriptive statistics of the families prior to characterising them in detail. This is a little bit confusing for the reader. Changing the order of the presented results could facilitate the understanding of the manuscript.

We presented the descriptive statistics of the families prior to characterising them in detail in order to avoid duplicate Tables 1-3, which are very long. The order of presented results follow a logical scheme: first we characterised the whole population of women and then we group women in communities, based on their characteristics (profiles 1-3). In our purposes, this should highlight the differences between the average-woman and the real-woman variability.

However, to make the reading more clear, and based on your suggestion, we changed the Table’s captions in this way:

Table 1. - Distribution of the characteristics included in the profile 1: overall population of women and focus on the families of community combinations statistically associated with BC. See paragraph 3.4 for the description of the communities and the families.

We believe all these changes have strongly improved our manuscript, as well as the bibliography upgrade and the English language revision.

Reviewer 3 Report

This i san interesting paper but I have concerns around clarity overall and potential (unaddressed) bias.

Abstract

Grammar throughout the abstract needs to be refined as the Engsh is sub-optimal

The flow does not make sense and should fist mention the rationale for breasts screening/or if rates are suboptimal (in other words why screening tests need to be maximised).

Line 29 The authors state: “Personalized preventative interventions are increasingly urgent to face the breast cancers increase”. This statement needs qualification (ie what breast cancer increase?) and the grammar also needs improved, suggest: “Personalized preventative interventions are increasingly urgently needed to address an increase in breast cancer”

Line 31: Grammar: “The true effects of individual choices are particularly hard to be defined” should read “The true effect of individual choices are particularly hard to define”

Line 37: Grammar “we combine the communities in risk profiles” should be “combined the communities in risk profiles”

Lien 37: The authors state: “Our results suggest that preventive programs focused on increasing physical activity should be widely promoted, in particular among the oldest women” however, they have not reported eg that physical activity if suboptimal and or clustered by community so the statement does not make sense

Line 42: The authors state: “Participatory data collection…could improve communities' self-empowerment” How do the authors come to this conclusion, again should be qualified

Introduction

The flow does not make sense…the authors first discuss healthcare/patient relationships then participatory research, then RCTs, then breast incidence (which should be qualified with statistics reported). They then go on to discuss quantification risk factor combination with no discussion of risk factors for breast cancer which should be to provide context t this statement. Recommend this part (with breast cancer incidence) is first then discussing why participatory research is important in relation to these issues.

Line 76 “Many risk models have undergone validation in study populations other than those used in initial development, or have been further assessed in comparative studies. Some models were developed based on hormonal and environmental factors that influence BC risk, while others rely more on family histories.” Need citations

Line 80, the authors state that “Comprehensive risk models that incorporate all known risk factors would im- 80 prove the ability to identify women at increased risk and promote risk-tailored screening”. I disagree – have the authors examined the PROCAS-1 and PROCAS-2. Prcs-1 examined current health, age at first period, age at first pregnancy, age at menopause, HRT use, diet, BMI and weight gain, and lifestyle choice to develop a risk score for breast cancer and subsequently involve risk stratified screening which the Manchester group on working on (with a far larger study sample). I suggest the authors need to quantify how their work is distinct form this work

Line 141 The authors state “confirm the current finding at the population level.” Which current finding? There is no citation

Line 145 The authors state: “Documentation burden and time were significantly associated with 145 a decrease in dedicated time for patient care, so patients’ personal history or behavioural 146 risk factors should be collected using a participatory approach”. What do the authors mean here? Isn’t provision of personal and family history by a patient participatory in its very nature?

Overall I found the introduction to be disjointed and no real rationale around why participatory methods have been discussed here as in general participatory methods imply participants are involved in codesign of research which is not the case here.

Methods

Measurement considerations: How were data on lifestyle collected? There is no mention of which questionnaires were used and if they are validated or this population. In addition, was breast cancer diagnosis verified as well as family history? In the absence of this being discussed I have serious concerns about the use of self-reported data and the bias it may have introduced into this study

Line 184 were the 5601 women selected on the basis of multiple screens? How they were selected needs to be made explicit as it is hard to judge if selection bias may be present

Line 212 the authors report: “A binomial logistic re- 212 regression model was also performed including all the variables that were statistically significant (p < 0.05) within the univariate analysis (Fig.1)” how was statistical significance derived?

Line 224 “each profile has been furtherly analyzed” should be “each profile was further analyzed”

I am not qualified to judge the network analysis

Results: Line 280, WCRF’s should be defined first time of use

Limitaiosn – Self reported data – there is no discussion around the limitations of this which is particularly important given there is also no information on whether the questionnaires used were validated or not

Discusis

Line 579 “Evidence from literature showed that healthy lifestyle behaviors tend to cluster”. Grammar should read: Evidence from literature shows that healthy lifestyle behaviors tend to cluster.

Line 591The authors state “a participatory approach by using a structured self-administered questionnaire investigating risk 592 factors, family history and lifestyle, to collect women’ personal information to be linked 593 with clinical data (histopathological proven BC and their features) provided by clinicians”. This is not a participatory approach, this is simply having research participants report their own data (which subsequently may be introducing measurement error that has not been acknowledged by the authors)

Line 614 The authors state: “the SNA show (GRAMMAR) that a better compliance to this recommendation is associated to a lower risk of BC onset, in particular among women belonging to other families 615 than the Fuchsia, Green and Yellow (Tab.3). It is also worth of note, that the FYF women 616 declaring a low compliance with the recommendation (Tab. B6 and Fig. 5) are positively 617 associated with BC” I am unsure how this is noteworthy (beyond groups of women in the same families displaying similar lifestyle behaviours, not unexpected) given postmenopausal breast cancer is significantly associated with obesity yet there is no mention of body weight in this discussion.

Conclusion The authors state: “a unique example of integrating a huge amount of clinical  data together with data on lifestyle and daily habits, to emphasize their crucial role in the development of BC, showing the importance of defining the influences of each single preventive/predisposing behavior and the effects of their combination to identify targeted  risk profiles”. Again I encourage the authors to look at the work around PROCAS-1 and PROCAS-2. Further, this clinical data is self-reported…

The authors also state: “In our opinion, participatory research and the scientific-based development of mobile health tools could be the winning strategies for promoting tailored prevention  policy and health education programs to improve communities' self-empowerment.” 1. This is not participatory research as already discussed and 2. There is no mention of mobile health tools anywhere else in the paper except in keywords and the conclusion as I am unclear on how the authors are able to draw these conclusions

Author Response

Dear Reviewer 3,

we thoroughly enjoyed your review and found your comments very precious for improving our work. We have made substantial changes to the manuscript according to your indications and to the comments of the other two reviewers. You will find them in the newly uploaded manuscript: the additions and deletions are in track changes mode. Bibliographic references are changed and improved, but to make the text more clear we have already accepted the revisions.

We completely changed the abstract, following your suggestions.

We thoroughly upgraded the introduction section following your suggested flow. We also better conceptualised the aims of the work, as you suggested (Lines 305-310) and we added a paragraph about the study limitations in the discussion section (Lines 851-864).

Below we provide point by point responses to your questions:

Abstract

We substantially changed all the abstract, following your suggestions. The updated abstract is a bit longer than the maximum number of words indicated in the authors guidelines.

Introduction:

We thoroughly restructured the introduction, following the suggested flow. We first introduced breast cancer incidence and its implications. We described the screening utility, introduced the risk-stratification approach and described the main BC risk factors. Then, we introduced the role that a community-based participatory approach could have in collecting data about specific personal features or behavioural risk factors in a more accurate way. To this purpose we also referred to the conclusions of the Procas studies (Lines 165-179).

We highlighted the hypothesis that the participatory research framework could help to overcome the current limitations in collecting data, as well as the use of electronic or mobile tools.

Finally, we better contextualised the aims of our current work: combining women’ personal features and lifestyle habits into profiles that could influence the BC risk and introducing an innovative tool, the Dress-PINK, designed to overcome the limitations of self-reported data in collecting population-based information in the BC field (Lines 305-310).

  • ex Line 76 : Many risk models have undergone validation in study populations other than those used in initial development, or have been further assessed in comparative studies. Some models were developed based on hormonal and environmental factors that influence BC risk, while others rely more on family histories.” we changed the statement and added citations. See Lines 93-98
  • ex Line 80: we referred to the Procas studies at Lines 165-179. As just mentioned, we used the conclusion of the Procas in a different way, compared to your suggestion. The PINK and the Procas studies have a basic difference in the nature of data: the Procas data concerned a population screening, while the PINK data concerns a self-selected cohort of women. As suggested by Reviewer 1, to compare women participating in the questionnaires and the general population we added some information extracted from the Eurostat database. In particular we refer to the Lines 443-446 within the Results paragraph 3.1. Additionally, we highlighted the selection bias limitation in the Discussion section.
  • ex Line 141: confirm the current finding at the population level.” Which current finding? There is no citation - we changed the statement
  • ex Line 145: Documentation burden and time were significantly [...] What do the authors mean here? Isn’t provision of personal and family history by a patient participatory in its very nature? - we changed the statement and discussed more in depth which information should be collected using other strategies than the physician interview to patients.
  • Overall I found the introduction to be disjointed and no real rationale around why participatory methods have been discussed here as in general participatory methods imply participants are involved in codesign of research which is not the case here.

We refer to community-based participatory approach as an orientation to research that focuses on relationships between research partners and goals of societal transformation, as promoted by Minkler & Wallerstein, 2003, rather than a specific set of research methods or techniques [Minkler M, Blackwell AG, Thompson M, Tamir H. Community-based participatory research: implications for public health funding. Am J Public Health. 2003;93(8):1210-1213. doi:10.2105/ajph.93.8.1210]. In our work, in particular, we consider the adjunct value of people participation in providing real-patient data, also using new technology as the mobile health tools.

 Methods:

  • Measurement considerations: How were data on lifestyle collected? There is no mention of which questionnaires were used and if they are validated for this population. In addition, was breast cancer diagnosis verified as well as family history? In the absence of this being discussed I have serious concerns about the use of self-reported data and the bias it may have introduced into this study.

Thank you for this comment that allowed us to describe more in depth the PINK questionnaire and the measurements. We added some parts in the material and methods section, to make the measurements characteristics clearer.

  • ex Line 184: were the 5601 women selected on the basis of multiple screens? How they were selected needs to be made explicit as it is hard to judge if selection bias may be present

(see Lines 318-366) First we better describe our population of recruited women and the selection process which reduced the number of analysed data, compared to the number of recruited women (29355).

In this work we only included women having available both the clinical and the self-reported data (10097 women: 5601 post-menopausal and 4446 premenopausal). Among the premenopausal patients the BC cases were 57 only. We believe this makes the application of SNA (and traditional statistical analysis) to premenopausal women too uncertain.

We also specified at Lines 335-338 that: The P.I.N.K. questionnaire items were jointly decided by the group of radiologists and epidemiologists participating in the study, based on a Delphi method and the questionnaire comprehension was tested among a random sample of 50 women of the general population.

Additionally, at the Lines 340-346, we indicated which information is collected by radiologists and which are self-reported. In particular:

  • BC diagnoses were provided by radiologists and were defined as a dichotomous variable (yes/not) using the anatomic pathology reports of BI-RADS 5 (B5) lesions resulting from needle core biopsies;
  • family history is self-reported by answering to these questions

-Has any of your first-degree relatives (mother, sister, daughter) developed breast or ovarian cancer? (Indicate the age of onset for each relative);

-Has any of your first-degree male relatives (father, brother, son) developed breast cancer? (Indicate the age of onset for each relative)

  • ex Line 212: A binomial logistic re- 212 regression model was also performed including all the variables that were statistically significant (p < 0.05) within the univariate analysis (Fig.1)” how was statistical significance derived?

(see Lines 389-393) we changed the original statement in the followingA binomial logistic regression model was also performed including all the variables that were statistically significant (p < 0.05) within the logistic regression univariate analyses (Fig.1). Each univariate analysis includes age and a single variable of the P.I.N.K questionnaire or breast density”.

  • ex Line 224: “each profile has been furtherly analyzed” should be “each profile was further analyzed” - done

 Results:

  • ex Line 280: WCRF was first defined at Line 130

Furthermore, self-reported data limitations were emphasised both within the introduction and the discussion.

 Discussion:

In accordance with your indications, we thoroughly restructured the discussion and the conclusion sections to make the flow of our purposes more consistent with the introduction and the aims of the work.

In particular, after reporting the limitations of the present study, we stressed the value of our results in illustrating a new method of analyses (the data driven approach) and for the development of the Dress-PINK.

  •  ex Line 579: Evidence from literature showed that healthy lifestyle behaviors tend to cluster”. Grammar should read: Evidence from literature shows that healthy lifestyle behaviors tend to cluster - done
  • ex Line 591: in this statement “participatory” were used in an improper way, so we deleted it
  • ex Line 614: “It is also worth noting that the FYF women….”. I am unsure how this is noteworthy (beyond groups of women in the same families displaying similar lifestyle behaviours, not unexpected) given postmenopausal breast cancer is significantly associated with obesity yet there is no mention of body weight in this discussion.

Please, see Lines 597-600 and 773-775: Within a family, combinations of communities share non-modifiable (profile 1) and personal history characteristics (profile 2) but differ in terms of lifestyle habits (profile 3). This is also evident looking at Table B7 (in Appendix B).

Concerning the body weight question, we did not discuss it for the following reasons:

  • As shown in Table B6, the FYF women have a bit higher rate of overweight and obese women (46.1% vs 43.1% of the Fuchsia family as a whole), but a lower rate if compared to the FYA women (52.9%), in particular referring to obesity. This last difference makes the association between BMI and each Fuchsia combination statistically significant (chi-square test p=0.011).
  • Additionally, the BMI distribution among the FYF women does not differ from that of the group of the other Fuchsia combinations taken as a whole (chi-square test p >0.05). Furthermore, BMI classes of the FYF women are not statistically associated with BC.

However, we added this statement at Lines 795-797:

Conversely, the FYF women do not statistically differ in terms of BMI or weight gain during menopause if compared to the group of all the other Fuchsia combination taken as a whole (Tab. 6B).

  • ex Line 614: the SNA shows (GRAMMAR) - We did not change the grammar because the verb “show” refers to the binomial logistic model and the SNA

Conclusion:

Conclusion The authors state: “a unique example of integrating a huge amount of clinical data together with data on lifestyle and daily habits, to emphasize their crucial role in the development of BC, showing the importance of defining the influences of each single preventive/predisposing behavior and the effects of their combination to identify targeted risk profiles”. Again, I encourage the authors to look at the work around PROCAS-1 and PROCAS-2. Further, this clinical data is self-reported…

According to your suggestions, we specified the meaning of “participation” in terms of “collaboration between stakeholders and researchers for building real-world evidence”.

We changed the “unique example” in “an example of integrating and analyzing in a new way a huge amount of clinical data together with data on lifestyle and daily habits, to emphasize their crucial role in the development of BC” see Lines 889-891.

As just mentioned we refer to the Procas studies at Lines 165-179.

We believe all these changes have strongly improved our manuscript, as well as the bibliography upgrade and the English language revision.

Round 2

Reviewer 1 Report

The paper is now ready for pubblication, the authors applied all the advices.

Many  compliments for the project

Reviewer 3 Report

I believe my review comments have been satisfactorily addressed.